# Targeting Histone Deacetylases: Opportunities for Cancer Treatment and Chemoprevention

**DOI:** 10.3390/pharmaceutics14010209

**Published:** 2022-01-16

**Authors:** Dusan Ruzic, Nemanja Djoković, Tatjana Srdić-Rajić, Cesar Echeverria, Katarina Nikolic, Juan F. Santibanez

**Affiliations:** 1Department of Pharmaceutical Chemistry, Faculty of Pharmacy, University of Belgrade, Vojvode Stepe 450, 11221 Belgrade, Serbia; dusan.ruzic@pharmacy.bg.ac.rs (D.R.); nemanja.djokovic@pharmacy.bg.ac.rs (N.D.); katarina.nikolic@pharmacy.bg.ac.rs (K.N.); 2Department of Experimental Oncology, Institute for Oncology and Radiology of Serbia, Pasterova 14, 11000 Belgrade, Serbia; tatjana.srdic@ncrc.ac.rs; 3Facultad de Medicina, Universidad de Atacama, Copayapu 485, Copiapo 1531772, Chile; cesar.echeverria@uda.cl; 4Group for Molecular Oncology, Institute for Medical Research, National Institute of the Republic of Serbia, University of Belgrade, Dr. Subotica 4, POB 102, 11129 Belgrade, Serbia; 5Centro Integrativo de Biología y Química Aplicada (CIBQA), Universidad Bernardo O’Higgins, Santiago 8370854, Chile

**Keywords:** histone deacetylases, cancer, epigenetic, chemoprevention, HDAC inhibitors, dietary-derived inhibitors, bifunctional inhibitors, PROTAC, clinical trials

## Abstract

The dysregulation of gene expression is a critical event involved in all steps of tumorigenesis. Aberrant histone and non-histone acetylation modifications of gene expression due to the abnormal activation of histone deacetylases (HDAC) have been reported in hematologic and solid types of cancer. In this sense, the cancer-associated epigenetic alterations are promising targets for anticancer therapy and chemoprevention. HDAC inhibitors (HDACi) induce histone hyperacetylation within target proteins, altering cell cycle and proliferation, cell differentiation, and the regulation of cell death programs. Over the last three decades, an increasing number of synthetic and naturally derived compounds, such as dietary-derived products, have been demonstrated to act as HDACi and have provided biological and molecular insights with regard to the role of HDAC in cancer. The first part of this review is focused on the biological roles of the Zinc-dependent HDAC family in malignant diseases. Accordingly, the small-molecules and natural products such as HDACi are described in terms of cancer therapy and chemoprevention. Furthermore, structural considerations are included to improve the HDACi selectivity and combinatory potential with other specific targeting agents in bifunctional inhibitors and proteolysis targeting chimeras. Additionally, clinical trials that combine HDACi with current therapies are discussed, which may open new avenues in terms of the feasibility of HDACi’s future clinical applications in precision cancer therapies.

## 1. Introduction

Numerous epigenetic alterations are recognized as hallmarks of cancer biology [1]. Changes in methylation, such as global hypomethylation and CpG island hypermethylation, along with acetylation, including hypoacetylation of histones H3 and H4 [2,3,4,5], have been identified in the early stages of carcinogenesis.

Specifically, histone proteins are core packaged with the eukaryotic DNA in a chromatin unit called the nucleosome. Nucleosomes comprise 146 base pairs of DNA wrapped by a histone octamer containing two H2A, H2B, H3, and H4 histones [6]. Chromatin can be in a condensed state related to transcriptional gene repression, while the decondensation of chromatin or open configuration permits the access of regulatory transcription factors to DNA and the control of RNA synthesis [7,8]. The regulation of chromatin configurations in its different active states is controlled through posttranslational modifications (PTMs) that primarily target amino acids within the N-terminal tail of the core histone proteins. PTMs encompass a wide range of chemical reactions, including ubiquitination, SUMOylation, GlcNAcylation, phosphorylation, methylation, and acetylation [9,10].

Histone acetylation status is controlled by reversible chromatin modifiers, such as histone deacetylases (HDACs), also called erasers, that remove an acetyl group (COCH_3_) on ε-amino groups of lysine residues in N-terminal regions. The deacetylation of histones results in chromatin condensation, or “heterochromatin”, leading to gene silencing. Histone hypoacetylation alters the net electrostatic charge on the histones and facilitates the interaction with the negative charge of DNA phosphate, which maintains chromatin in a condensed structure [11,12]. Contrary to the deacetylating reaction, the accumulation of acetyl-lysine residues on histones is mediated via histone acetyltransferases (HATs), also called writers. As the positive charge of acetylated histones neutralizes the interaction with the negatively charged DNA, a chromatin decondensation occurs towards a “euchromatin” configuration, which allows the accessibility to transcription factors and active gene transcription (Figure 1) [7,13].

Although cancer implicates multistep processes with a series of transformations related to oncogenes’ activation and tumor suppressors’ inhibition, it is currently known that epigenetic dysregulation highly influences cancer initiation and progression and tumor plasticity and heterogeneity [14,15,16]. Alterations in the epigenetic control of the chromatin status and histones during cellular transformation largely influence cellular homeostatic control and cancer cell proliferation; these modifications result from the several DNA mutagenic events occurring during tumor progression [17,18]. Among epigenetic dysregulation events in cancer progression, dysfunctions of the HDACs and the related modifications of the protein acetylation levels highly influence cancer cells’ malignancy [19]. In this sense, HDACs, due to their importance in epigenetic abnormalities, have been developed as important therapeutic targets to control cancer progression and malignancy.

Aberrant HDAC expressions are involved in multiple different stages of cancer and become one of the hallmarks in hematological malignancies and solid tumors [20,21,22]. Additionally, the increased expression of HDACs may be associated with poor outcomes and advanced disease in cancer patients, such as gastric, ovarian, neuroblastoma cancer, and multiple myeloma, among others [22]. As crucial players in cancer, HDACs are involved in the regulation of several cellular and molecular events (Figure 2). HDACs influence cell cycle and cellular proliferation, and HDAC inhibition induces cell cycle arrest at the G1 phase by reducing the expression of cyclins and cyclin-dependent kinases (CDK) or inducing the expression of CDK inhibitors [23,24]; HDACs inhibition also regulates cancer cell apoptosis via the regulation of the expression of pro- and antiapoptotic factors, such as cell surface death receptors and/or ligands including FAS/APO1-FASL, TNF-TNF receptors and TRAIL-TRAIL receptors, reduction in the expression levels of cytoplasmic FLICE-like inhibitory protein (c-FLIP), Bax, and Bcl2 family members [25,26]. Likewise, DNA-damage repair (DDR) is critically regulated by HDACs due to its role in modulating chromatin reorganization, maintaining the dynamic equilibrium of acetylation of DNA-damage repair proteins, as well as influencing almost all events in DNA repair such as base excision repair, nucleotide excision repair, and mismatch repair [27]; Autophagy, with demonstrated involvement in the development, maintenance, and progression of cancer, is susceptible to regulation by HDACs. In cancer cells, HDACs deacetylate cytoplasmic proteins involved in the regulation of crucial autophagy proteins such as LC3-II and Beclin1, as well as the intracellular signaling proteins mTor, apoptosis-inducing factor (AIF), and p53 [28,29,30,31,32,33]. Epithelial-to-mesenchymal transition (EMT) is a crucial process in cancer cell invasion and metastasis. The EMT implicates an extensive range of the silencing of epithelial genes and the activation of mesenchymal genes via transcriptional mechanisms that directly implicate histone modifications. For instance, transcriptional repressors of the E-cadherin (CDH1), such as Snail or ZEB1, which recruit HDACs to CDH1 promoter for histone deacetylation, result in gene silencing and potentiate the EMT [34,35]. As tumor mass reaches 1–2 mm in size, the angiogenic process is activated for the supply of oxygen and nutrients and the disposal of carbon dioxide and waste [36]. Angiogenesis is tightly regulated by the hypoxic microenvironment and the transcription factor hypoxia-inducible factors-1 α (HIF-1α). HDACs either directly deacetylate HIF-1α and inhibit the protein degradation or indirectly facilitate HIF-1α stabilization by deacetylating its chaperones, HSP70 and HSP90 [37,38].

This review will focus on the first generation of classic small-molecule and dietary-derived HDAC inhibitors with demonstrated anticancer and chemopreventive activities. Additionally, we will discuss some perspectives of addressing new ways of improving the HDACi selectivity and the combinatory potential of HDACi, and other specific targets that could be included in future clinical applications.

## 2. Histone Deacetylases Overview

The discovery of this first mammalian histone deacetylase (HDAC1) marked the beginning of the HDACs family. HDAC research dates from 1964, when a relationship between the acetylation status of histones was linked to RNA synthesis, followed by the description of a putative enzyme with histone deacetylase activity in 1969 [39,40]. Nonetheless, the first HDAC was discovered in 1999 using an affinity matrix based on the covalent HDAC inhibitor trapoxin B. The sequencing of an isolated 46 kDa protein with deacetylase enzymatic activity revealed a resemblance with the yeast transcriptional regulator encoded by the RPD3 gene [41].

Eleven HDAC isoforms have been described and grouped into four classes based on their homology with yeast deacetylases (Table 1) [42,43]. Class I encompasses HDAC1, 2, 3, and 8 with sequence homology to yeast deacetylase reduced potassium dependency-3 (Rpd3), which are mainly found in the nucleus and are part of multiprotein complexes that are included in the epigenetic control of gene expression, cell proliferation, and survival. Meanwhile, class II comprises HDAC4, 5, 6, 7, 9, and 10, sharing sequence homologies with the yeast histone deacetylases-1 (Hda1). HDAC4, 5, 7, and 9 are classified into a class IIa group and are found shuttling between the nucleus and cytoplasm. HDACs 6 and 10 are categorized as class IIb due to a second catalytic domain and are primarily found in the cytoplasm. Moreover, class IIa enzymes form multiprotein complexes with relatively weak deacetylase activity, while class IIb HDACs are active deacetylases. Class IV includes a unique and the smallest member, HDAC11 isoform, mainly found in the nucleus, which shares sequence identity with Rpd3 and Hda1 yeast deacetylases [12,44,45,46,47]. Additionally, these 11 HDACs are characterized by a Zn^2+^ in their active sites, which is fundamental for hydrolase activity, and known as the zinc-dependent “classical” HDACs [48]. The class III group encompasses seven NAD^+^-dependent histone deacetylase Sirtuins that are not in the scope of this review [42].

HDAcs are promising targets for cancer chemoprevention and chemotherapy depending on their involvement in gene expression and the variety of cellular functions involved in carcinogenesis, cell growth, survival, and homologous recombination [49,50,51]. Moreover, the manipulation of histone acetylation through HDAC inhibition has been proposed as a mechanism for derepressing genes dysregulated in chronic conditions such as neurodegenerative diseases, psychological disorders, and cancer [52,53].

## 3. Histone Deacetylases Inhibitors in the Treatment of Cancer

For decades cancer has been defined as an autonomous disease governed by the mutational activation of oncogenes and/or the inactivation of tumor suppressor genes, which triggers a sequence of molecular events that in time favor tumorigenesis [54]. Reasonably, chemotherapy’s primary and ultimate aim is to eliminate malignant cells and restore cancer patients’ cellular and physiological homeostasis. Historically, chemotherapy is based on cytotoxic drugs to target the highly proliferating cancer cells potentially vulnerable due to the aberrant molecular mechanisms associated with malignant transformation [54]. Without a doubt, chemotherapy is one of the most significant advances to treat cancer patients. It exerts unpredictable or undesirable toxic effects that target normal cells, partly because therapeutic drugs do not discriminate between rapidly dividing cancer cells from highly proliferative non-malignant cells. Many efforts have been made to overcome the lack of specificity of anticancer drugs; in this sense, in recent years, the concept of targeted therapy or precision oncology has been developed to design therapeutic molecules targeting the potential cancer-specific molecular and cellular signals. Therefore, the new era of cancer therapies considers mapping crucial genes and signaling pathway components and the development of ‘smart’ drugs that surgically target cancer cells, reducing the nonspecific effects on normal cells [55,56,57].

In cancer, epigenetic events crucially contribute to tumor development and malignant progression. Tumorigenesis implicates multiple events beyond genetic alterations, such as epigenetic regulation that influences gene expression and protein activities, making epigenetic targeting an attractive therapeutic strategy to fight cancer [55,58].

As mentioned above, HDAC inhibitors promote proliferation arrest and differentiation, trigger the cell death program, and control the tumor angiogenesis immune responses necessary for tumor growth and spreading and escape from immune surveillance. In this sense, the “epigenetic vulnerability of cancer cells” hypothesis has been coined; consequently, HDACs could be essential for the expression of crucial genes implicated in the survival and growth of transformed cells, providing a relative specificity of HDACs inhibition as opposed to normal cells [59,60]. The rationality of the use of drugs considers that epigenetic alterations are reversible; thereby, anti-epigenetic compounds may in some way recover the non-transformed scenery. Meanwhile, chromatin regulators are susceptible to oncogenic mutations, offering attractive molecular targets for cancer treatment [60,61]. The following sections will address the main HDACi according to their chemical structure and impact on cancer treatment.

### 3.1. Histone Deacetylase Inhibitors

Besides mediating the acetylation status of various proteins, the zinc-dependent HDACs possess a mostly conserved catalytic active site [62,63]. To effectively inhibit the hydrolysis of acetyl-lysine residues on the histone tails and non-histone proteins, HDACi drugs should possess a cap group, aliphatic or aromatic linker, and a zinc-binding group (ZBG), as a part of a generic (classical) pharmacophore model (Figure 3) [64,65]. The ZBG includes carboxylic acid, hydroxamic acid, 2-aminobenzamide, thiol, and others, and it is crucial for designing an HDACi. The zinc-chelating group is inserted with the active site of the HDACs and mainly by bidentate mode interacts with and sequesters the zinc ion. When positioned in the substrate-binding tunnel, the linker domain mimics the lysine residue and establishes hydrophobic interactions with the amino acid residues in the crevice of the HDAC active site.

Furthermore, the cap is usually built from a heterocyclic or carbocyclic group that interacts with the aromatic amino acid residues located either close to the outer domain of the active site or at the external surface of the enzyme [64,66]. This cap group is susceptible to chemical modifications to enhance the selectivity of the HDACi. Nevertheless, recently extended pharmacophore models have been proposed that can include the majority of currently known HDAC inhibitors and may help classify and design new selective HDAC inhibitors [67,68].

Chemically, HDACi can be classified according to their ZBG (Table 2): aliphatic (short-chain fatty) acids; hydroxamic acids; ortho-aminoanilides, also called benzamides; thiols and cyclic peptides [55,69]. In general, aliphatic acids seem to be the weaker HDACi inhibitor with some limited class I HDAC selectivity because they weakly bind to Zn^2+^ ions inside the catalytic pocket of the HDAC isoforms. The concentrations of carboxylic acid HDACi required to induce histone hyperacetylation are usually between micromolar and millimolar. Contrary to carboxylic acids, hydroxamic acids inhibit a broad range of HDACs (1–11) with nM potency, and ortho-aminoanilides, and cyclic peptides are highly selective for class I HDACs [70].

#### 3.1.1. Aliphatic Acids

Sodium butyrate is produced by intestinal microbiota from dietary fibers and selectively inhibits all class I HDAC isoenzymes [70,71]. Sodium butyrate is capless HDACi with carboxylic acid as ZBG, and it has been studied for almost 40 years as a chemopreventive agent [71]. Butyrate was demonstrated to function as a weak ligand-competitive HDACi with a Ki of 46 μM when tested in MCF-7 breast cancer cell lysates in vitro, compared with the Ki of 1 nM for trichostatin A (TSA) in similar experimental conditions [72]. This small-molecule HDACi induces the expression of the cyclin-dependent kinase inhibitor 1 (p21/WAF1) expression, which suppresses cell-cycle progression at the G1 phase and induces cellular differentiation [73]. Interestingly, butyrate is an effective natural product that acts on primary and secondary chemoprevention in model systems of colon cancer [74]. Sodium phenylbutyrate has a similar HDAC inhibitory profile as sodium butyrate, and it was studied as differentiation therapy in certain malignant diseases [75].

Valproic acid (2-propylpentanoic acid), a well-known anticonvulsant drug, inhibits class I HDAC isoforms, and it is currently being used in sixty clinical trials in patients with different neoplastic diseases [76]. The epigenetic molecular mechanisms mediated by valproic acid include the hyperacetylation of histones H3 and H4, which leads to growth inhibition, the induction of differentiation, erythroid maturation, and the inhibition of metastasis formation [77,78].

#### 3.1.2. Hydroxamic Acids

Microbial metabolites provide many of the first HDACi. In this sense, the Streptomyces hygroscopicus metabolite (TSA), described by Yoshida thirty years ago, remains one of the most potent available HDACi [52,79]. In 1990, TSA was described to induce the in vitro accumulation of highly acetylated histones, given the first evidence of the inhibition of HDAC activity [79]. TSA is a pan-HDAC inhibitor that structurally consists of a substituted phenyl ring cap group, a conjugated rigid diene linker region that sits in the substrate channel, and a hydroxamic acid tail functioning as a bidentate zinc chelator in the active site of HDACs. The general pharmacophore of HDAC inhibitors (zinc-binding group, linker, and cap group) is illustrated in Figure 3.

Moreover, TSA can be considered a structural precursor prototypical inhibitor that served as a model to design novel hydroxamate derivatives such as HDAC [66,80,81]. Although TSA initially was described as a non-competitive HDAC inhibitor, later reports that reanalyzed the enzyme kinetics showed that TSA acts as a competitive inhibitor [71,79]. Besides, TSA widely affected the tumor cells’ gene expression and was demonstrated to be useful as a therapeutic drug in other diseases, including asthma and neurodegenerative conditions [82,83,84].

Vorinostat (suberoylanilide hydroxamic acid, SAHA) is a linear HDACi structurally and functionally related to TSA; it chelates the Zn^2+^ with a hydroxamic acid moiety in a bidentate mode, as was demonstrated in studies of the vorinostat crystal structure within the active site of bacterium *A. aeolicus* HDAC1 homolog [85]. This drug was the first drug approved by USFDA in 2006 for cutaneous T-cell lymphoma/leukemia (CTCL) that inhibits class I, II, and IV HDACs [86]. When vorinostat is applied with synthetic triterpenoids, the formation of estrogen receptor-negative mammary tumors can be successfully inhibited in the mouse model [87]. As numerous SAHA derivatives are synthesized to date, selenium-based SAHA derivatives retain HDAC inhibitory activity and prevent the development of melanoma in laboratory-generated skin reconstructs [88]. Although several toxicity reactions of vorinostat/SAHA are reported during clinical trials, the single use of suberoylanilide hydroxamic acid is limited in primary chemoprevention.

To explore the possibilities of modulating the histone acetylome in cancer therapies, HDAC inhibitors with high affinity for DNA were synthesized [89,90]. Belinostat (also known as PXD101) inhibits HDACs in a nanomolar range (IC_50_ of 27 nM) [91]. It was the third agent approved in 2014 by USFDA for relapsed or refractory peripheral T-cell lymphoma (PTCL), which inhibits class I and II HDACs with nanomolar potency [92,93].

Panobinostat (LBH589 or Faridak^®^) is a pan-HDACi, cinnamic hydroxamate with potent antitumor activity in preclinical models and promising clinical efficacy in cancer patients. Panobinostat was approved in 2015 by USFDA for multiple myeloma therapies due to its capacity to inhibit class I, II, and IV HDACs [94,95]. In preclinical models, panobinostat was demonstrated to prevent the development of N-methylnitrosourea-induced rat mammary tumors and inhibit lung tumorigenesis induced by 4-(methylnitrosamino)-1-(3-pyridyl)-1-butanone [96,97]. Additionally, panobinostat demonstrates relevant anti-tumorigenic activities in lymphoid malignancies, ovarian cancer, and pancreatic cancer [98,99,100]. Moreover, in combination with resveratrol (RVT), panobinostat can further enhance the proapoptotic activities of Sirtuin1 on malignant lymphoid cells [101,102].

Ricolinostat (ACY-1215) also belongs to the class of hydroxamate HDAC inhibitors, in which the hydroxamate moiety coordinates with Zn^2+^ in bidentate mode, forming a canonical five-membered chelate complex with Zn^2+^ of HDAC6 [103], which potently and selectively inhibits HDAC6 (IC_50_ value of 5nM), and is 12-, 10-, and 11-fold less active against HDAC1, HDAC2, and HDAC3, respectively. Meanwhile, it has minimal activity (IC_50_ > 1 μM) against HDAC4, HDAC5, HDAC7, HDAC9, HDAC11, and has slight activity against HDAC8 (IC_50_ = 0.1 μM) [104]. Ricolinostat has demonstrated acceptable tolerability and anti-myeloma efficacy upon combination treatment with lenalidomide and dexamethasone, as well as pharmacodynamic evidence of both HDAC6 and class I HDAC inhibition in multiple myeloma patients [105]. Furthermore, ricolinostat demonstrated a favorable safety profile in patients with relapsed and refractory lymphoid malignancies in a Phase I/II clinical trial (NCT02091063) [106]. In preclinical studies, ricolinostat efficiently reduced tumor burden and increased survival in a mouse xenograft melanoma model [107]. Furthermore, ACY-1215 enhances the anticancer activities of oxaliplatin in colorectal cancer [108]. In addition, the HDAC6 inhibition provokes increased cellular microtubule stability induced by cold and nocodazole treatment due to hyperacetylation of the α-tubulin, which suppresses the microtubule dynamic instability in cancer cells [109].

Citarinostat (ACY-241) was designed as a second-generation orally available HDAC6 selective inhibitor (IC_50_ 2,6 nM) with improved solubility properties over the structurally related inhibitor ricolinostat. Citarinostat, in combination with paclitaxel, significantly suppresses solid tumor growth in xenograft models [110].

#### 3.1.3. Benzamides

Entinostat (MS-275-SNDX-275) is a synthetic benzamide derivative HDACi, which potently and selectively inhibits class I and IV HDAC enzymes. It is an orally bioavailable drug with a moderate variability in exposure and exerts cell proliferation inhibition, which induces terminal cellular differentiation and apoptosis. Its selective inhibitory activity on class I isoform HDACs suggests a potential better safety and efficacy than nonselective pan-HDAC inhibitors. It has been included in several clinical trials addressing advanced types of breast cancer and non-small lung cancer, among others [111,112,113,114].

Chidamide (CS055/HBI-8000; Tucidinostat), an analog of the class I HDACi entinostat, is a novel HDAC low nanomolar inhibitor of the benzamide class, which specifically inhibits HDAC1, 2, 3, and 10. It was approved by the China FDA and EMA (European Medicines Agency) to treat hematologic malignancies [115,116,117].

Recently, an HDAC 1 and -2-selective entinostat derivative, MPT0L184, was described to trigger premature mitosis with potential use to counteract cancer therapy resistance. Structurally, MPT0L184 comprises a substituted isostere of the purine ring, namely 7H-pyrrolo[2,3-d]pyrimidine, at the cap region, and the N-benzyl linker binds to a 2-aminoanilide moiety [118].

Mocetinostat (MGCD0103) is a selective inhibitor of class I and IV HDACs (isoforms 1, 2, 3, and 11) that does not affect the class II HDACs and was developed by MethylGene of Canada [119]. It is an orally bioavailable agent with significant antitumor activity in vivo against a broad spectrum of human cancer types, and antitumor activity is achieved at clinically achievable doses [120]. Moreover, in preclinical human cancer cells models, MGCD0103 induces core histone H3 and H4 acetylation at micromolar doses. Additionally, mocetinostat, administered with gemcitabine, seems a valuable therapeutic strategy to reverse chemoresistance in gemcitabine-resistant metastatic leiomyosarcoma patients [121]. Additionally, this drug has been studied in phase 2 clinical trials for refractory lymphoma therapy with an acceptable safety profile [122].

#### 3.1.4. Cyclic Peptides

Romidepsin (also called FK228, FR901228, depsipeptide, or Istodax^®^) was the second drug approved by USFDA in 2009 for cutaneous T-cell lymphoma (CTCL), and in 2011 for the therapy of peripheral T-cell lymphoma (PTCL), and is currently undergoing clinical trials for the treatment of other malignancies [123,124]. Romidepsin is a cyclic tetrapeptide isolated from the fermentation product of Chromobacterium violaceum, developed by Gloucester Pharmaceuticals. It is comprising L-valine, D-valine, (Z)-dehydrobutyrine, D-cysteine, and (3S)-hydroxy-7-mercapto-4-heptenoic acid, connected by an internal disulfide bond generating the bicyclic structure [125,126]. Interestingly, romidepsin is found as a pro-drug after intracellular reduction of the disulfide bond by glutathione, allowing the active free dithiol to attach to the active site and chelate the Zn^2+^ in class I HDAC in a monodentate way [48,127]. Romidepsin potently inhibits HDACs with IC_50_ values for HDAC 1 of 36 nM and HDAC2 of 47 nM and provokes cell growth inhibition and apoptosis induction in leukemic cells [48,127,128,129,130]. This drug has been approved as second-line therapy for treating cutaneous T-cell lymphoma (TCL) and peripheral TCL [131].

### 3.2. Natural Compounds in Targeting Histone Deacetylases for Cancer Chemoprevention

Chemoprevention can be defined as the use of natural or synthetic chemical agents to prevent, revoke, or reverse tumorigenesis or tumor progression in order to reduce the inherent risks associated with cancer [132,133]. It is believed that preventing cancer from developing to the advanced malignant stages is a promising strategy in controlling cancer mortality due to the limited treatment in the last steps of tumorigenesis [133,134].

Because of the complexity of cancer, chemopreventive agents are expected to act, at the molecular level, in the three steps of carcinogenesis: initiation, promotion, and progression [135]. For instance, chemopreventive agents that act at the initiation stage are essential for DNA preservation in its native form by blocking DNA mutagenesis. These compounds may inactivate carcinogens and prevent irreparable DNA damage. Additionally, they can facilitate carcinogen metabolization and thus improve the anti-oxidative system by acting as free-radical scavengers, either by themselves or by enhancing anti-oxidative enzyme activities [135,136]. At the same time, chemopreventive compounds that influence cancer promotion and progression inhibit the proliferative capacity of initiated cells, in part, by acting on intracellular mitogenic signals such as NF-κB and mTOR, among others [137,138,139]. Additionally, agents acting in these steps are expected to reduce or delay metastasis by promoting apoptosis, and inhibiting angiogenesis, the EMT, cell invasion, and the colonization of distant organs [135].

Natural bioactive compounds have been investigated intensely for their potential chemopreventive activities in cancer. Dietary-derived agents exhibit a significant advantage in chemoprevention, as they generally have no or low adverse effects compared with the long-term administration of pharmaceutical drugs [134,135].

Moreover, about 60–70% of chemotherapeutic drugs are natural compounds or serve as a chemical scaffold for new drug synthesis and have contributed considerably to pharmacotherapy [140,141]. In addition, natural products have diverse pharmacological activities and are also crucial in providing novel lead templates for drug discovery and development. Furthermore, natural products’ biological effects rely partly on their capacity to bind bio-macromolecules that somehow have been improved in nature [142,143]. Dietary factors provide many new opportunities to develop new chemopreventive agents with potent anticancer activities [12]. In this sense, dietary compounds may also serve as a pharmacological core for new drugs to reprogram cancer cells to “normal” differentiation features and regulate gene expression by targeting the cancer epigenetic dysregulation [144,145,146,147]. Currently, due to the rich source of active dietary compounds, the pursuit of new, more specific, and more potent HDAC inhibitors has moved towards dietary factors; this aims to identify novel drugs with HDACs inhibition activities for cancer treatment and other epigenetic dysbalance-associated diseases, such as fragile X syndrome and autoimmune disorders [148,149,150]. Next, selected natural compounds describing prominent HDAC inhibition and anti-cancer and chemopreventive functions are described (Table 3).

Among dietary-derived HDAC inhibitors, n-butyrate was one of the first compounds identified due to its capacity to provoke a reversible histone hyperacetylation in transformed cells [151]. Moreover, n-butyrate exhibits a chemopreventive potential on colon cancer [152].

Similarly, dietary allyl derivative or organosulfur compounds from garlic were described to influence the acetylation status of histone in neoplastic cells [153]. For instance, allyl mercaptan (AM) and diallyl disulfide (DADS) were shown to possess HDAC inhibitory properties and demonstrated a comparatively more potent inhibition of HDAC8 in cell-free conditions [154]. Under treatment with either AM or DADS, cancer cells show increased acetylation levels of histones H3 and H4 associated with the tumor suppressor CDK inhibitor 1A (*CDKN1A*) gene promoter. The upregulation of *CDKN1A* expression promotes p21^cip1/waf1^ overexpression, which subsequently provokes cell proliferation arrest [155,156,157].

Isothiocyanates (ITC), generated from the hydrolysis of brassica or cruciferous vegetable glucosinolates, exert an anticancer function through HDACs inhibition [146,158,159]. Benzyl isothiocyanate (BITC) was demonstrated to inhibit HDAC enzymatic activities and downregulate HDAC1 and HDAC3 expression, which may function as an NF-κB inactivation mechanism resulting in the growth suppression of pancreatic cancer cells [160]. In turn, sulforaphane (SFN) inhibits HDACs, such as HDAC6, in cancer colon cells alongside histone hyperacetylation, cell proliferation arrest, and cellular apoptosis, whereas in prostate cancer models, SFN reduces HDAC1, HDAC4, and HDAC7 expression levels [161,162].

Phenethyl isothiocyanate (PEITC) is shown to have promising chemopreventive activities due to its HDAC inhibitory functions in several cancer models [163,164,165,166]. PEITC inhibits the binding of HDACs to the euchromatin in tumor cells and induces a de-repression of the tumor suppressor p21^Cip1/WAF1^ expression in prostate cancer cells [164,167]. In addition, indole-3-carbinol (I3C) and its derivate product of acid condensation, 3,3′-diindolylmethane (DIM), have demonstrated potential chemoprotective properties. DIM has been effective in downregulating the expression of class I HDAC1, HDAC2, and HDAC3 proteins, resulting in the inhibition of the antiapoptotic protein survivin expression and increased P21 and P27 expression in cancer cells [48,165,168,169,170].

Furthermore, a growing number of dietary-derived flavonoids have been identified for HDAC’s inhibition, which possesses anticancer and anti-inflammatory functions [171,172].

Quercetin (3,5,7,3′,4′-pentahydroxyflavone) is a ubiquitous and abundant flavonoid in plants and fruits that inhibits HDAC1 and HDAC8 in cancer cells [173,174,175]. Apigenin (5,7,4′-trihydroxyflavone) is a phytoestrogen aglycone abundantly found in plants belonging to the *Asteraceae* family genera *Artemisia*, *Achillea*, *Matricaria*, and *Tanacetum* [176]. Apigenin induces the inhibition of HDAC1 and HDAC3 alongside increasing histones acetylation on gene promoters of p21^Cip1/WAF1^ and subsequent apoptosis [177,178]. Chrysin (5,7-dihydroxyflavone) is a naturally occurring flavone belonging to a group of polyphenolic compounds found in mushrooms, olive oil, tea, red wine, and passion fruit flowers, as well as Thai propolis and honey, which exerts antitumor activities on various cancer cell types [48,179,180]. Chrysin inhibits HDAC8 activity and reduces melanoma cells’ HDAC-2, 3, and 8 protein levels. Moreover, it induces cell cycle arrest at the G_1_ phase and chromatin remodeling at the p21^CIP1/WAF1^ promoter gene due to increased histones H3 and H4 hyperacetylation [181].

Curcumin (diferuloylmethane) is a bioactive lipophilic polyphenol isolated from the rhizome of *Curcuma longa* of the *Zingiberaceae* (Ginger) plant family. The chemopreventive activity of curcumin, in part, lies in its capacity to modulate the epigenetic regulation of cancer cells [182,183,184]. Curcumin inhibits HDAC4 and HDAC6 in medulloblastoma and leukemic cells, respectively, while increasing histone acetylation on gene promoters of the proapoptotic BAX gene due to inhibition of HDAC1, 3, and 8 activity and expression in leukemic cells [185,186,187].

(−)Epigallocatechin-3-gallate (EGCG) is the more abundant polyphenol of green tea that can also be found in curry spices, grapes, soy, and berries [188]. Among the chemopreventive activities, EGCG can inhibit class I HDAC1, HDAC2, and HDAC3 and increase histone acetylation, leading to cell cycle arrest and apoptosis in cancer cells [189,190,191,192].

Resveratrol (3,5,4′-trihydroxytrans-stilbene) is a naturally occurring nonflavonoid stilbene polyphenol compound in grapes and wine, with cancer chemoprevention properties, because of its multitargeting activities on cancer initiation, promotion, and progression [182,193,194]. RVT is a pan-inhibitor of all eleven human HDACs of class I and II and is associated with the inhibition proliferation of cancer cells [195,196].

Genistein (5,7,4′-trihydroxyisoflavone) is a naturally occurring compound that structurally belongs to isoflavones and is found profusely in soybeans [197]. Genistein exhibits a wide range of biological properties, such as antioxidant, anti-inflammatory, antiangiogenic, proapoptotic, and antiproliferative activities, which explain the chemopreventive and therapeutic potential of this isoflavone [198]. In cancer cells, genistein stimulates the expression of the tumor suppressor genes p21^WAF1^ and p16^INK4a^ associated with reducing HDACs activity in breast cancer and colon cancer [199,200]. Moreover, Genistein inhibits HDAC1, HDAC5, and HDAC6 in colon, prostate, and human cervical carcinoma cells [201,202,203].

## 4. Perspectives in the Development of New HDAC Inhibitors

To date, approved HDACi for clinical treatments, such as vorinostat, romidepsin, belinostat, and panobinostat, exhibit relatively low specificity. For instance, as was abovementioned, vorinostat inhibits class I, II, and IV HDACs, romidepsin inhibits predominantly class I HDACs, belinostat inhibits class I and II HDACs, and panobinostat class I, II, and IV HDACs. Moreover, all drugs are indicated mainly for the treatment of hematological cancers with shallow success in treating solid tumors. Although the underlying reason is not well understood, the low capacity of the agents to reach within the solid tumor and potential HDACi in vivo instability may influence the success of the treatments [204,205,206,207]. Additionally, the common mechanisms within HDACs classes, such as zinc-dependent functions, may impose a barrier to developing anti-HDACs agents with highly specific activities.

### 4.1. Structural Considerations to Design Histone Deacetylases Inhibitors

Recently, a new expanded model for HDAC classical pharmacophore has been proposed, with the aim to consider more possibilities for the design of more specific and selective HDAC inhibitors. Besides the ZBG, linker, and cap group, Melesina et al. [208] proposed that the catalytic pocket of HDACs can be composed of two main entities: the main pocket that comprises the acetate binding cavity, the substrate-binding tunnel, and the rim of the pocket; and the sub-pockets, such as the side pocket (SP), a lower pocket (LP), and the foot pocket (FP) (review in [209]). Additionally, it is crucial to consider that class IIb HDACs possess a pseudo-catalytic domain or a unique zinc-finger domain as in HDAC6. The classical model perfectly fits with inhibitors targeting the main pocket but do not fully explain the features of inhibitors that also interact with sub-pockets domains, as is observed in the case of TH65 targeting the SP of HDAC8, cyclopropyl hydroxamic acid derivative that addresses the LP of HDAC4, and p-thienyl-anilinobenzamide derivative that targets the of HDAC2 [208,210,211]. Therefore, this expanded model may significantly contribute to designing new, more potent, and selective HDACs inhibitors.

Accordingly, strategies to design selective HDAC inhibitors implicate an optimization of all three HDAC pharmacophoric features. The ZBG is present in most HDAC inhibitors. For instance, the hydroxamic acid ZBG fits all HDAC isoforms but is less favorable for class IIa HDACs, while trifluoromethyloxadiazole ZBG is preferred by class IIa HDACs [212,213]. In contrast, *ortho*-aminoanilide inhibitors exhibit selectivity towards HDAC1, 2, and 3 [214]. Despite the high conservation of the amino acid residues around the zinc ion among HDAC isoforms, the selectivity of different ZBGs might be achieved due to the structural differences in the acetate binding cavity [213]. The second strategy which can be implemented to obtain isoform-selective HDAC inhibitors is the modification of the linker attached to the ZBG and placed in the substrate-binding tunnel. It is possible to identify pan-HDAC interacting linkers and selective HDAC interacting linker domains. For instance, the hydroxamic acid TSA and the cinnamic acid derivatives belinostat and panobinostat, with a vinyl linker region, and vorinostat with saturated alkyl linkers, exhibit high inhibitory potency and a poor HDAC isoform selectivity, respectively [98,215,216,217,218]. In comparison, inhibitors with aromatic linkers possess high isoform selectivity toward HDAC6, HDAC8, and HDAC10 depending on the cap group [209]. For instance, benzohydroxamic acid exhibits an IC_50_ of 115 nM on HDAC6 and selectivity between 17- and 290-fold greater than the other HDACs. At the same time, cyclohexene hydroxamic acid possesses an IC_50_ of 12 nM on HDAC6 and a selectivity about 36- to 760-fold greater than other HDACs. Moreover, both are small and capless compounds with significantly HDAC selectivity [219]. For tubacin, a highly potent and selective HDAC6 inhibitor (IC_50_  =  4 nM), the selectivity is due to specific interactions between the unique capping motif and the surface topology of HDAC6 [219,220]. Thus, in general, it is essential to define the nature of the linker region in compound design, as aliphatic linkers are nonselective in comparison with the aromatic and carbocyclic rings often preferred by HDAC6 [209]. Another strategy is the modification of the cap group, which interacts with the edge of the binding pocket and the surface of the protein. Although the protein margin area’s structural diversity offers many possibilities for inhibitors to adopt bioactive conformations, it can be challenging to predict a ligand’s binding conformation. Regardless, some cap residues exhibit class I HDAC selectivity. For example, the natural cyclic depsipeptides, such as romidepsin and largazole, and cyclic tetrapeptides, such as HC toxin and chlamydocin, possess HDAC1 selectivity over HDAC6 [221,222,223]. Not only the nature of the cap influences the HDACi selectivity, but the position of the cap group may also improve the isoform specificity of inhibitory agents [209]. This condition is mainly observed in benzohydroxamic acids with a meta-substitution at the aromatic linker to generate HDAC inhibitors with improved selectivity on HDAC8 over HDAC1 and HDAC6 [224]. Similarly, benzohydroxamic acids with an N-substituted indolyl-6-hydroxamic acid core exhibit 290-fold greater selectivity against HDAC8 than HDAC1-3 and HDAC6 [225]. Other strategies consider the optimization of the lower-pocket-targeting group and the foot-pocket-targeting group [209].

### 4.2. Bifunctional Histone Deacetylase Inhibitors

In preclinical models, HDACi monotherapy has been shown to be effective for hematological and solid tumor treatments; these compounds seem to be well tolerated and with low toxicity to normal tissues, while in clinical trials, the success, especially on solid tumors, is minimal, and secondary effects are observed [226]. Therefore, new combinatory treatments have been explored to resolve the issues of HDAC monotherapies in cancer patients. The combinatory strategies with current anticancer agents seem to work in synergistic or additive antitumor modes, which substantially improve the conventional therapies due to the enhancement of therapeutic efficacy, including reduced side effects of HDACi [227].

In this sense, new treatments have been developed by conjugating two distinct therapeutic compounds in a single molecule for dual-targeting strategies [228,229,230]. The use of bifunctional drugs may provide increased efficacy by targeting additional disease-related pathways, mitigate side effects, reverse drug resistance by blocking specific mechanisms of resistance, offer the advantage of a more predictable pharmacokinetic profile, modulate drug resistance, and reduce patient compliance difficulties and clinical trial costs, as well as drug–drug interaction adversities [231].

Among the different linked mechanisms in creating a multitarget ligand, HDAC-based dual inhibitors are synthesized through a pharmacophore fusion approach; the secondary targeting agents are mainly linked to the cap of the selected single-targeting HDAC inhibitors (Table 4) [64,232]. Dual HDAC inhibitors are considered as a combination with a variety of molecular targets. For instance, class I HDAC/kinase dual inhibitors have been developed. The bifunctional inhibitor CUDC-101, resulting from the fusion of hydroxamic acid-based HDACi with the methoxyethoxy group of the phenylamino quinazoline backbone of the receptor tyrosine kinase (RTK) inhibitors, potently inhibits HDAC and EGFR and HER2 with IC_50_ = 4.4 nM, 2.4 nM, and 15.7 nM, respectively [233]. CUD-101 demonstrated a strong anticancer function in anaplastic thyroid cancer, pancreatic cancer, and glioblastoma, among others [234,235,236].

HDAC–tubulin dual inhibitors also display improved antitumor activities. In this sense, the combination of colchicine with hydroxamic acid and *o*-aminoanilide moieties generated compounds with potent class I HDAC inhibition, with particular selectivity for HDAC2 (IC_50_ = 0.19 µM) over HDACs 1 and 3 (IC_50_ = 1.50 and 1.49 µM, respectively) for one of the compounds (6a). Moreover, this hybrid compound retained similar activities to colchicine. Additionally, a second dual inhibitor compound (11a), although it exhibits lower potency against HDACs 1–3 (IC_50_ = 12.50, 6.73, and 11.23 µM, respectively), shows a better antiproliferative activity in a panel of 11 human cancer cell lines than the first compound and colchicine [237].

Class II HDAC6-selective dual inhibitors designed to simultaneously target Janus kinases (JAKs) via the conjugation of a JAK inhibitor with a hydroxamic acid ZBG have been developed. Specifically, the JAK2-selective inhibitor pacritinib conjugated with Vorinostat/SAHA generated a series of hybrids incorporating aromatic or nonaromatic linkers of varying lengths and either a monodentate carboxylic acid or bidentate hydroxamic acid ZBG. One of the compounds demonstrated a selective JAK2/HDAC6 dual inhibitory capacity with an IC50 value of 2.1 nM for HDAC6 and the most significant JAK2 inhibition, among all hybrids, with an IC_50_ = 1.4 nM. Moreover, a high HDAC6 selectivity was displayed, with a selectivity >1000-fold greater than HDAC3 (IC_50_ = 2.17 µM), >100-fold greater than HDACs 1,8 and 11, and >20-fold greater than HDACs 2 and 10, and slight activity against the class IIa isoforms. In addition, this bifunctional inhibitor shows a >50-fold greater selectivity for JAK2 over the other JAK isoforms [238].

Interestingly, dual anti-epigenetic compounds have also been developed. HDAC1 or HDAC-2 form a complex with scaffold protein REST corepressor 1 (RCOR1/Co-REST) alongside the epigenetic eraser lysine-specific demethylase 1 (LSD1); LSD1 catalyzes the removal of activating mono- and dimethyl marks on nucleosomal H3K4 residues, resulting in transcriptional repression. Additionally, a critical functional interplay between HDACs and LSD1 within the complex has been reported, since the enzymes mutually influence each other’s activity through the CoREST protein [239]. The conjugation of the LSD1 inhibitor tranylcypromine to various *o*-aminoanilide and hydroxamic acid-based HDAC inhibitors generated a variety of bifunctional HDAC and LSD1 inhibitors. One of these hybrid compounds (7), resulting from the conjugation with vorinostat/SAHA, exhibits the most potent inhibitory activity against HDAC1 and HDAC2, with IC_50_ of 15 nM and 23 nM, as well as potent inhibition against LSD1, with IC_50_ of 1.20 μM. Moreover, this compound shows more robust antiproliferative activities than vorinostat/SAHA against several cancer cell lines in vitro [240].

### 4.3. Proteolysis Targeting Chimeras

One characteristic of small-inhibitor compounds is that they are designed, for example, to inhibit the catalytic activities of the protein target, while the potential non-catalytic functions are not affected. A new method of drug design implicates the complete protein target depletion by constructing proteolysis-targeting chimeras (PROTACs) that achieve the regulation of both enzymatic and non-enzymatic functions [241].

Specifically, PROTACs regulate the expression of the protein of interest at the post-translational level by directing the targets to the cellular proteolytic machinery, such as the ubiquitin–proteasome pathway [242]. PROTAC technology makes druggable the traditional “unable to medicine” targets and has tremendous potential in overcoming drug resistance [241,243]. PROTACs comprise heterobifunctional compounds engineered to recruit the protein target to the E3 ubiquitin ligase for ubiquitination, and subsequently proteasomal degradation [244,245]. The unique PROTACs mechanism of action provides several advantages, such as a catalytic mode of action instead of high equilibrium target occupancy, improving potency, enhancing selectivity compared with the initial promiscuous targeting warhead, persistent target protein depletion, and potential tissue–cellular selectivity, compared with conventional small-compound inhibitors [241]. Furthermore, because PROTACs exert an acute and reversible post-translational protein depletion, this technology offers better advantages than other genetic knockdown methodologies. Additionally, PROTACs may function as molecular probes to elucidate the epigenetic regulation in health and diseases as a new modality in precision medicine [246].

In 2018, one of the first HDAC-targeting PROTACs was described as the first HDAC6 degrader, generated based on nonselective HDAC inhibitors with E3 ubiquitin ligase ligand pomalidomide as cereblon (CRBN) (Table 5). CRBN ligands rely on the structure of the anticancer drug thalidomide and its derivatives [247]. The PROTAC compound exhibits a degrader concentration (DC)_50_ of 34 nM and a maximum percentage of HDAC6 degradation (D_max_) of 70.5% without significant effects on other HDACs family members. Moreover, the degradation of HDAC6 results in an upregulation of tubulin acetylation in treated MCF-7 cells [248]. Similarly, An et al. [249] synthesized an HDAC6 degrader (NP8) by combining Next-A, at the end of the aliphatic chain, with the CRBN pomalidomide. NP8 provokes a significant inhibition of the proliferation of the multiple myeloma cell line MM.1S, comparable with parental agent Nex-A, with the DC_50_ value for NP8 of 3.8 nM. Moreover, the HDAC6-pomalidomide degrader showed a high grade of specificity, as other representative HDAC family members were not affected by NP8 treatment. Additionally, Yang et al. [250] described the synthesis of a degrader (NH2) combining Next-A, at the NH2 residue, with pomalidomide as CRBN. NH2 effectively depletes HDAC6 in MM.1S with a DC50 of 3,2 nM after 24 h of treatment; moreover, the HDAC6–NH2–CRBN PROTAC provokes HDAC6 degradation at 1 h of treatment, reaching maximum degradation at around 6–8 h. In addition, the NH2 washout in HeLa cells recovered the HDAC6 protein level within 3 h, confirming the suitability of use of NH2 as a reversible knockdown compound for HDAC6 protein in cancer cells.

In addition, a cell-permeable HDAC-targeting PROTAC based on Von Hippel-Lindau (VHL) as an E3 ligase-binding motif and Nexturastat A (Next-A), a well-known HDAC6-selective inhibitor, has been synthesized. VHL has demonstrated efficacy and the robust degradation of a wide range of protein targets and shown the ability to bypass the possible off-target effects of CRBN recruitment [247]. The resulting PROTAC agent induces HDAC6 degradation with a DC_50_ of 7.1 nM and a D_max_ of 90% in the MM1S cell line [251]. Additionally, SR-3558, a potent and selective inhibitor of class I HDACs, with VHL as an E3 ligase ligand, generates a selective HDAC3 degrader (XZ9002) with a DC_50_ value of 42 nM [252]. Similarly, Smalley et al. [253] generated a PROTAC molecule targeting the nucleus-localized class I HDACs. The class I *o*-aminoanilide HDAC inhibitor CI-994 was conjugated to the VHL ligand via a twelve-carbon alkyl linker to synthesize a PROTAC with a DC_50_ of 1 µM for HDACs 1, 2 and 3 in HCT116 human colon cancer cells.

Finally, Cao et al. used bestatin, a cellular inhibitor of apoptosis (cIAP) ligand, as E3 ligase-binding motifs to develop cIAP–HDAC–PROTACs [254]. Namely, cIAP ligands function as E3 ubiquitin ligases that target RIP1 for polyubiquitination and have shown success in the design of PROTACs [247,255]. Several HDAC–cIAP hybrid molecules were synthesized and one in particular, named P1, significantly reduced the HDAC1, HDAC6, and HDAC8 expression levels on the B lymphocytes cell line RPMI-8226 after 24 h treatment, at concentrations ranging from 1 μM to 4 μM, although it did not induce HDAC degradation at 6 h of treatment [255].

## 5. Combined Clinical Strategies with Histone Deacetylases Inhibitors

The clinical approval of vorinostat, romidepsin, belinostat, panobinostat, and tucidinostat/chidamide allows the inclusion of HDACi as part of the therapeutic armament for an increasing number of cancers. Nevertheless, HDACi, when used as a mono-therapeutic chemical agent, showed a limited cancer application, mainly in treating hematological malignancies. Moreover, tumor resistance to HDACis treatment has been observed alongside a limited therapy achievement of solid tumors [256,257], whereas combining HDACis with chemotherapy agents may improve and maximize cancer treatment and reduce treatment-associated toxicity side effects due to reduced agent doses. Furthermore, combined therapies can synergistically avoid or reduce undesirable cancer treatment resistance [257]. In this sense, HDACis may function as chemosensitizers and synergistically work with an increasing number of diverse therapeutic chemical drugs, biological-derived active proteins and polypeptides, and immunotherapies [258]. Thus, combined therapeutic strategies of HDACis with other approved cancer therapies may increase the efficacy of clinical approaches and have shown great promise in clinical trials. A considerable number of clinical trials have been implemented toward the combined application of HDACis, and in this section we will analyze selected clinical studies focusing mainly on solid tumors (Table 6).

Receptor tyrosine kinases (RTKs), due to regulating a vast range of biological and molecular functions and their dysregulated intracellular signals in several cancers, have been shown to be promising onco-targets. In particular, RTKs promote cancer cell proliferation, increasing c-Myc and cyclin D1 oncogenes expression. Cyclin D1 interacts with class I/II HDACs to regulate c-Myc expression, thus making the combinations of RTK targeted therapies and HDACi important strategies for cancer therapy [258,259,260,261]. A phase I/II clinical trial (NCT01027676), addressing the capacity of vorinostat to synergize and overcome resistance to the EGFR-TKI gefitinib in treated patients with advanced non-small-cell lung cancer (NSCLC), has demonstrated this combination to be feasible and well tolerated at the clinical regime of biweekly 400 mg/day vorinostat and daily doses of 250 mg gefitinib. EGFR-mutant NSCLC-treated patients showed a median progression-free survival (PFS) of 9.1 months and median overall survival (OS) of 24.1 months. This combination showed potential for improving the efficacy of gefitinib in EGFR-mutant NSCLC and suggested the potential benefit of vorinostat for improving the efficacy of EGFR-TKIs in these patients [262].

Angiogenesis is one of the hallmark events contributing to tumor growth and metastasis [36]. For instance, HDACs stabilize and inhibit the degradation of HIF-1α [37,38]. Meanwhile, the vascular endothelial growth factor (VEGF) secreted by cancer cells stimulates endothelial cells to form new blood vessels. Moreover, anti-VEGF treatment in preclinical studies has demonstrated the additive and synergistical benefits of combined treatments with other onco-therapies [263]. In this sense, a clinical study has tested the safety and efficacy of vorinostat and the VEGF blocker bevacizumab in metastatic clear-cell renal cell carcinoma (ccRCC) patients that were previously undertreated with different agents, such as sunitinib, sorafenib, axitinib, and temsirolimus, among others. A total of 200 mg vorinostat twice per day, for 14 days, and bevacizumab 15 mg kg^−1^, intravenously, every 21 days were administered as phase II doses. The toxicity of combined therapy seems to be acceptable and safe, and the median PFS and OS were 5.7 months and 12.9 months, respectively. Nevertheless, the study is limited due to the sample size and the single-arm design [264].

Additionally, hormone signaling is dysregulated in several cancers, such as breast and prostate cancer, and contributes to promoting tumorigenesis [258,265]. Thus, combined hormonal therapies with HDACis are promising strategies for the treatment of hormone-dependent cancers. Recently, a phase III trial of tucidinostat/chidamide (30 mg twice a week for four weeks) combined with the steroidal aromatase inhibitor exemestane (25 mg orally daily) improved the PFS compared with placebo plus exemestane, with a median PFS of 7.4 months and 3.8 months, respectively. Moreover, a substantially increased overall response and clinical benefit was observed in combined therapy of patients with advanced, hormone receptor-positive (HR^+^) and HER2-negative breast cancer that progressed after prior endocrine therapy [266]. Another phase I clinical study explored the safety and preliminary efficacy of combined entinostat therapy (3 and 5 mg orally per week) with the hormonal therapy drug enzalutamide in castration-resistant PCa (CRPC) patients. The median duration of treatment with entinostat was 18 weeks. Entinostat did not affect the steady plasma concentration of enzalutamide, and no defining dose-limiting toxicity (DLT) related to entinostat in these patients was observed. Despite the small number of enrolled patients (6), 5 mg of entinostat weekly combined with enzalutamide seems to have an acceptable safety profile [267].

In addition, a multicenter, randomized, double-blind, placebo-controlled phase III study (E2112) enrolled patients with advanced HR-positive, HER2-negative breast cancer whose disease progressed after nonsteroidal aromatase inhibitors had been performed. This phase III clinical trial was based on a previous ENCORE301 phase II study that reported improvement in PFS and OS with combined entinostat and exemestane therapy [268]. In this study, the patients were randomly treated with oral 25 mg exemestane once daily and 5 mg entinostat or placebo 5 mg once weekly. The results indicate that the median PFS of exemestane plus entinostat showed no significant difference with 3.3 months and 3.1 months, respectively.

Similarly, no median OS differences between exemestane plus entinostat and exemestane plus placebo arms were obtained, being 23.4 months and 21.7 months, respectively. Although the pharmacodynamic endpoint analysis indicated that in the exemestane plus entinostat arm a significant increase in lysine acetylation in PBMCs by C1D15 analysis was observed, confirming HDACi function, the combination of entinostat and exemestane did not improve advanced endocrine-resistant breast cancer patients’ outcomes [269].

In addition, HDACs regulate autophagy signals, including mTor pathways, that help tumor cells survive under metabolic stress conditions [22]. Moreover, preclinical approaches identified the mTOR/HDAC inhibitor combination as a promising strategy for cancer treatment [270]. In this regard, a phase I dose-finding trial (NCT01582009) for the mTOR inhibitor everolimus combined with the HDAC inhibitor panobinostat in advanced clear-cell renal cell carcinoma (ccRCC) patients has been completed. Based on the maximum tolerated dose, the therapeutic strategy was everolimus 5 mg PO daily and panobinostat 10 mg PO 3 times weekly (weeks 1 and 2) given in 21-day cycles. The 6-month PFS was 31%, with a median of 4.1 months. Although the combined everolimus/panobinostat therapy seems to be safe and tolerable in the indicated dosing regimen, with myelosuppression as the major DLT, this pairing did not appear to improve the clinical outcomes in the studied group of advanced ccRCC patients [271].

To date, chemotherapy continues to be among the most widely used cancer treatments, causing cell death. Despite the initial therapeutic success, cancer cells often develop chemoresistance, particularly in non-germ-cell solid tumors, which severely reduce the clinical benefits for cancer patients [272]. Additionally, HDACi therapies, due to the broad role of HDAC in diverse cellular functions, have been demonstrated to act as sensitizer agents for chemotherapy [273]. For instance, combining cisplatin with HDACi shows a promising strategy for increasing the efficacy of this platinum-based agent. A phase II, single-arm clinical trial was conducted to investigate the efficacy and safety of a combination treatment of tucidinostat/chidamide with cisplatin in metastatic advanced triple-negative breast cancer (TNBC) patients. Women with TNBC were treated with tucidinostat/chidamide and cisplatin, 20 mg twice weekly for 2 weeks, and 75 mg/m^2^ on a 21-day cycle, respectively. Despite the fact that this trial had several limitations, including a small sample size, non-randomized character, and the un-selection of the enrolled patients, the combination of chidamide and cisplatin did not seem to exhibit a superior efficacy compared with classical cisplatin-based chemotherapy in treating advanced TNBC [274]. Additionally, romidepsin has been included in a phase I dose-escalation study (NCT01902225) in combination with liposomal doxorubicin, an anthracycline that intercalates within DNA base pairs and inhibits topoisomerase II to treat cutaneous T-cell lymphoma (CTCL) and peripheral T-cell lymphoma (PTCL). Liposomal doxorubicin dose was fixed at 20 mg/m^2^ i.v. over 1 h on day 1 of each 28-day cycle, and the maximum tolerated dose (MTD) of romidepsin was determined to be 12 mg/m^2^. The results indicate that the overall response rate (ORR) in the CTCL patients cohort was 70%, while the PTCL cohort was 27%. Combined romidepsin and liposomal doxorubicin therapy regimes indicated high and rapid response rates in CTCL patients. The authors concluded that drug combination is highly active, safe, and well tolerated [275].

Furthermore, preclinical studies demonstrated the feasibility and efficacy of the combination of DNA synthesis inhibitor agents and HDACi in improving cancer sensitivity [276]. An open-label, non-randomized phase I/II clinical trial study (NCT00372437) addressed the safety and efficacy of the class I/IV HDACi mocetinostat combined with DNA synthesis inhibitor gemcitabine in patients with solid tumors. The therapeutic regime included gemcitabine at 1000 mg/m^2^, day 1 of three consecutive weeks, in 4-week cycles, and 50–110 mg, three times per week of oral mocetinostat. MTD and the recommended phase II dose for mocetinostat was 90 mg, while the ORR was 11% in the phase I study, and non-responses were observed in phase II. Due to the lack of significant positive results and significant toxicities in patients with advanced pancreatic cancer following the combined therapy, the study was terminated without merit for further testing [277].

One characteristic of cancer cells is their capacity to regulate immune functions and escape from immune surveillance [278]. In turn, HDACi demonstrated the ability to promote cancer cells’ immunogenicity [279], making the combination of HDACi with immunotherapies promising for cancer treatment. For example, the combination of vorinostat and the anti-PD-1 pembrolizumab, an immune checkpoint inhibitor, was evaluated in a phase II study (NCT02538510) in recurrent/metastatic squamous cell carcinomas of the head and neck (HN) and salivary gland cancer (SGC) patients. The therapeutic regime was conducted with oral 400 mg vorinostat 5 days on and 2 days off during each 21-day cycle, while intravenous pembrolizumab 200 mg was administrated every 21 days. The combined HDACi and immunotherapy demonstrated activity in HN patients with fewer responses in SGC. Although the combination demonstrated activity in HN, with fewer responses in SGC, high-grade toxicities compared with single-agent pembrolizumab were observed. Nevertheless, the obtained data provide new clues for the activity and tolerability of the proposed combined HDACis and immunotherapies in HN and SGC patients [280].

Similarly, a PEMDAC phase II clinical trial (NCT02697630) combining entinostat and pembrolizumab in patients with metastatic uveal melanoma (UM) showed manageable toxicities and no treatment-related patient deaths. Patients were treated intravenously with 200 mg pembrolizumab every third week in combination with entinostat 5 mg orally once a week. Median OS was 13.4 months, and OS at one year was 59%, while median PFS was 2.1 months, and one-year PFS was 17%. Additionally, combined therapy can cause tumor regression in a small subset of patients with metastatic UM, which provides data for a small subset of patients who can benefit from combined epigenetic therapy and immunotherapy [281]. Furthermore, a phase Ib clinical trial combining the second-generation HDAC6-selective inhibitor citarinostat (ACY-241) with the immune checkpoint inhibitor nivolumab has been conducted in patients with advanced NSCLC. Patients received orally administered citarinostat once daily for 28 days of two 28-day treatment cycles with a modified Fibonacci sequence of 180 mg QD, 360 mg QD, and 480 mg QD. For cycle 1, nivolumab was administered on day 15 at 3 mg/kg by intravenous injection, while in cycle 2, nivolumab was administered every 2 weeks on days 1 and 15. The results of this single-arm and dose-escalation study indicated an MTD for citarinostat of 360 mg. Additionally, a transient increase in histone (class I target) and tubulin (HDAC6 target) acetylation levels following treatment were observed in patients’ samples. Together, the results indicate that a combined HDACi and immunotherapy strategy is safe and tolerable and may be feasible in patients with advanced NSCLC [282].

Finally, two-phase I studies (NCT01755975 and NCT02341014) have combined romidepsin with the immunomodulatory drug lenalidomide and proteasome inhibitor carfilzomib in relapsed/refractory lymphoma to evaluate the MTD. Two study regimes were performed, A: romidepsin and lenalidomide, and B: and romidepsin, lenalidomide, and carfilzomib. The clinical trial included the enrollment of T-cell lymphoma (TCL) and B-cell lymphoma (BCL) as separate cohorts. The results indicate an MTD for regime A of romidepsin 14 mg/m^2^ IV on days 1, 8, and 15 and lenalidomide 25 mg oral on days 1–21 of a 28-day cycle, while for regime B an MTD of romidepsin 8 mg/m^2^ on days 1 and 8, lenalidomide 10 mg oral on days 1–14 and carfilzomib 36 mg/m^2^ IV on days 1 and 8 of a 21-day cycle. The ORR for regime A was 50% TCL and 47% BCL with a PFS of 5.7 months. The ORR was 50% TCL, 50% BCL with a PFS of 3.4 months for regime B. The results demonstrate that both regimes exhibit activity and acceptable safety profiles in both subtypes of relapsed/refractory lymphoma patients [283].

## 6. Conclusions and Concluding Remarks

Cancer is characterized by dysregulation in the balance of protein acetylation due to aberrant alterations in the expression and activity of HDACs, making epigenetic alterations one of the hallmarks of cancer generation and progression to metastatic stages. In this sense, HDACs targeting has emerged as a promising strategy in cancer treatment. To date, five HDACi have been approved for clinical approaches: three hydroxamic acids (vorinostat, panobinostat, and belinostat), one cyclic peptide (romidepsin), and one benzamide (chidamide). The clinical use of these drugs strongly supports the importance of targeting HDACs in cancer therapy and opens the path to designing new compounds with better efficacy, high selectivity, and lower side effects. Mainly, HDACi demonstrated efficacy for treating hematological malignancies and lymphomas. At the same time, an increasing number of clinical trials are ongoing to evaluate the safety and therapeutic benefits for refractory, advanced, and recurrent solid tumors. Unfortunately, most clinical trials revealed adverse issues, such as different grades of thrombocytopenia, neutropenia, anemia, arrhythmia, and gastrointestinal toxicity. At molecular levels, the adverse effects are alongside several events, such as abnormal gene transcription, genomic instability, and aberrant free radical production, which reduce the therapeutic success of HDACi [284]. Therefore, the design of new HDACi to overcome undesirable side effects and increased efficacy to penetrate solid tumors represents a critical pharmacology challenge.

In addition, a deeper understanding of the molecular mechanisms of the anticancer functions of HDACi, analyses of HDAC structural conformations, and chemical-specific modifications are necessary to improve the therapeutic efficacy of HDACi. This review describes HDACs’ features and their involvement in several cancer-associated cellular and molecular events, which picture the HDAC involvement in cancer physiopathology. Two main groups of HDACi inhibitors are analyzed according to their use for cancer treatment and dietary-derived compounds for chemoprevention. However, they can be highly intertwined since natural agents provide almost unlimited chemical structures for HDACi design.

In addition, the currently approved drugs such as HDACi for cancer treatment displayed relatively low selectivity. Due to the diversity of HDACs and cellular-specific functions, the design of more potent isoform-selective inhibitors is highly needed. According to the pharmacophore model, HDACis are susceptible to be modified in all three of the main groups (ZBG, linker, and cap), which may help design selective HDAC inhibitors via the optimization of the interaction with particular HDACs. Additionally, due to the oncogenic nature of cancer and the development of more potent and selective oncogene inhibitors, new investigations which addressed the potential of bifunctional inhibitors, which involve oncoprotein inhibitors and specific HDACi, have shown the potential of improving conventional therapies, in part by enhancing therapeutic efficacy and reducing HDACi side effects. Furthermore, PROTAC technology helps in the generation of a new class of HDAC targeting agents. PROTAC leads to the proteasomal degradation of HDACs with several advantages, such as target protein depletion and potential tissue-cellular selectivity that, with the development of more HDACi specificity, may highly improve cancer treatment and drug resistance. Finally, the HDACi can be combined with different basic anticancer experimental strategies, for inhibiting tumor growth, angiogenesis, metastasis, and apoptosis induction, which has allowed the execution of an increasing number of combined clinical trials. Together, these therapeutic strategies might benefit the survival and improve the quality of life of cancer patients when traditional treatments fail.

## Figures and Tables

**Figure 1 pharmaceutics-14-00209-f001:**
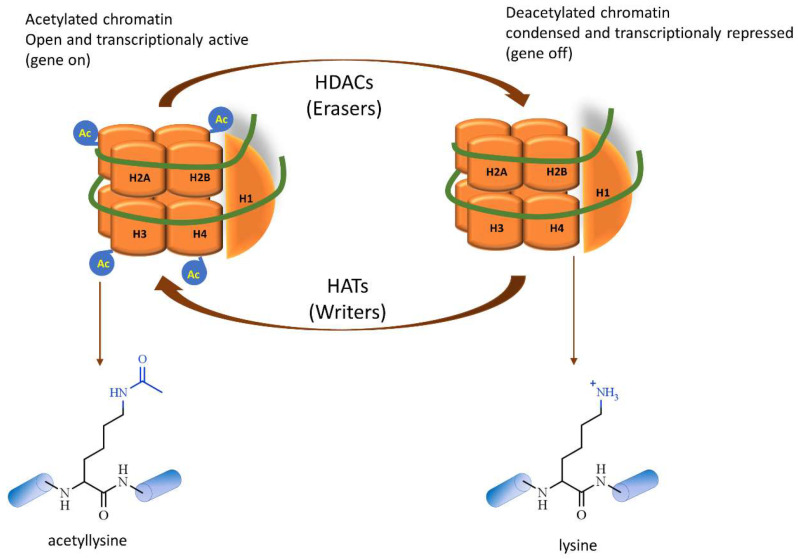
Overview of acetylation/deacetylation of histone lysine residues. The acetylation of lysine (Lys) residues catalyzed by histone acetyltransferases (HATs) induces decondensed chromatin and transcriptionally active DNA. In contrast, the histone deacetylases (HDACs) remove acetyl (Ac) residues and provoke condensed chromatin and repression of DNA transcription.

**Figure 2 pharmaceutics-14-00209-f002:**
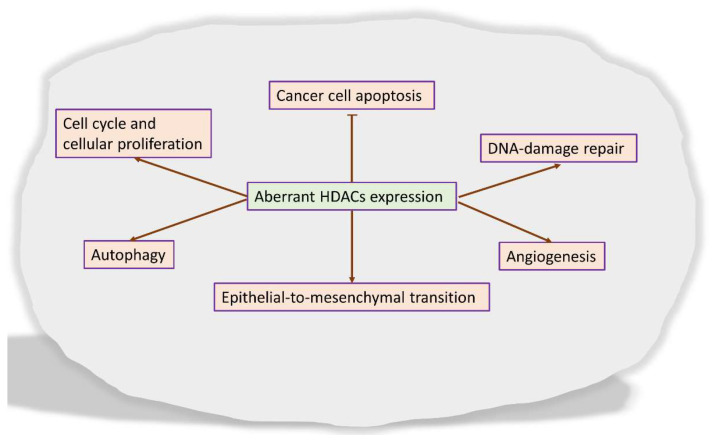
The hallmarks of histone deacetylases in cancer biology. HDACs enzymes mediate cancer cell malignancy by promoting cell proliferation, autophagy, DNA-damage repair, and tumor angiogenesis while inhibiting apoptosis. Furthermore, HDACs contribute to the expression of metastatic phenotypes by triggering the epithelial to the mesenchymal transition program. Arrows indicate stimulation, and the bar represents inhibition.

**Figure 3 pharmaceutics-14-00209-f003:**
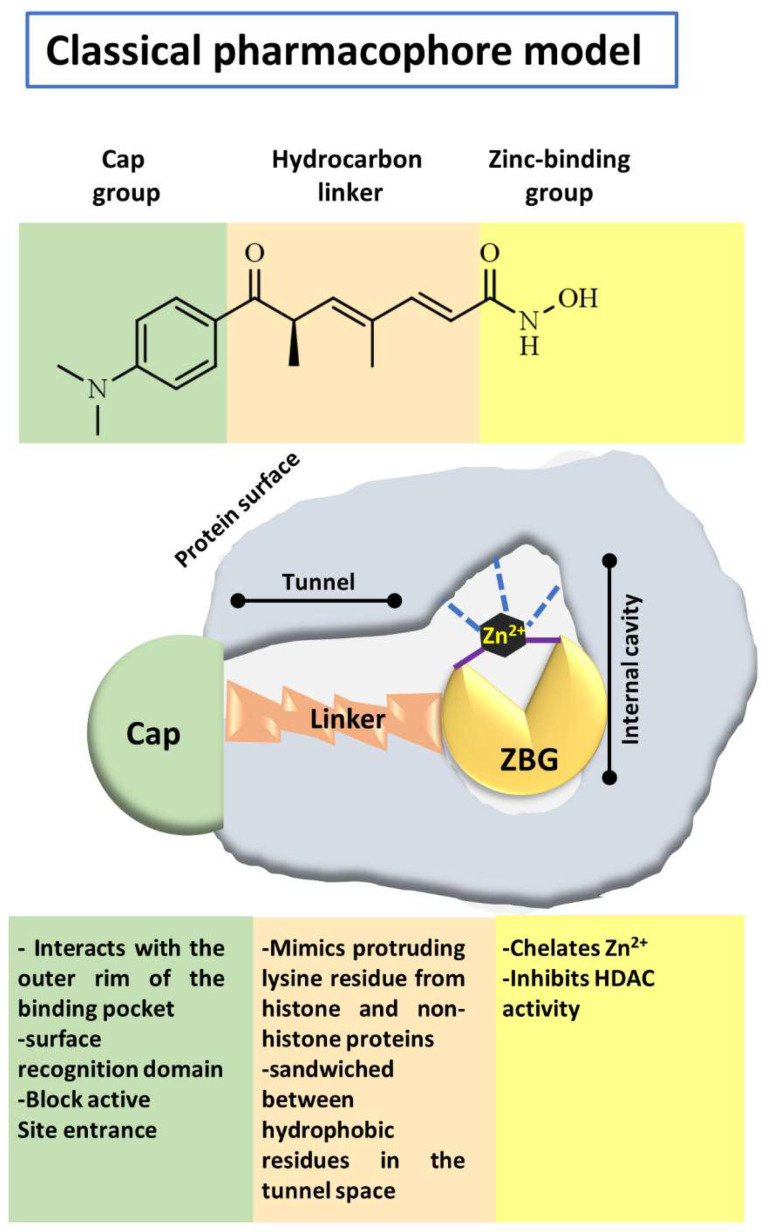
Classical pharmacophore model of histone deacetylases inhibitors. The classical pharmacophore model of HDAC inhibitors (HDACi) considers three motifs and is illustrated with trichostatin A agent: a hydrophobic cap group (green) that participates in the protein recognition and interaction, a hydrocarbon linker (magenta), and a hydrophilic domain that interact with the Zinc cation (Zn^2+^) at the enzyme active site called Zinc-binding group (ZBG) (yellow).

**Table 1 pharmaceutics-14-00209-t001:** Main cellular localization and basic molecular features of HDACs are indicated.

Histone Deacetylases
Class	Member	Cellular Localization	ChromosomePosition	Aminoacids N^o^(Molecular Weight, kD)	Basic Structure
I	HDAC1	Nucleus	1p35-p35.1	483 (51)	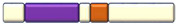
HDAC2	6q21	488 (55)	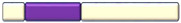
HDAC3	5q31.3	428 (49)	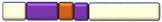
HDAC8	Xq13.1	377 (42)	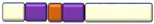
IIa	HDAC4	Nucleus/cytoplasm	2q37.3	1084 (119)	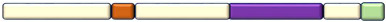
HDAC5	17q21.31	1122 (122)	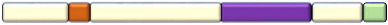
HDAC7	12q13.11	912 (103)	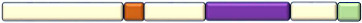
HDAC9	7p21	1069 (118)	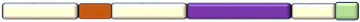
IIb	HDAC6	Cytoplasm	Xp11.23	1215 (131)	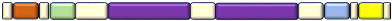
HDAC10	2q13.33	669 (71)	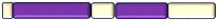
IV	HDAC11	Nucleus	3p25.1	343 (39)	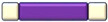

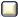
 N and C terminal regions, 
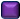
 Catalytic domains, 
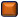
 Nuclear localization sequence, 
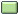
 Nuclear export sequence, 
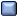
 cytoplasmic anchoring motif, 
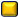
 Zinc finger motif.

**Table 2 pharmaceutics-14-00209-t002:** Selected HDAC inhibitors.

Histone Deacetylases Inhibitors				
Clasification	Name	HDACs (IC_50_)	Structure	Ref.
Aliphatic carboxylic acids	Sodium butyrate	HDAC1 (16 mM), HDAC2 (12 μM), HDAC3 (9 μM), HDAC8 (15 μM)	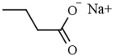	[72]
	Valproic acid	HDAC1 (38 mM), HDAC2 (62 mM), HDAC3 (161 μM), HDAC8 (103 μM)	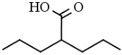	[76,77,78]
Hydroxamic acids	Vorinostat	HDAC1 (30 nM), HDAC2 (144 nM), HDAC3 (6 nM), HDAC6, (10 nM) HDAC8 (38 nM), HDAC10 (21 nM), HDAC11 (28 nM)	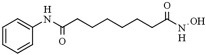	[85,86]
	Belinostat	HDAC1 (41 nM), HDAC2 (125 nM), HDAC3 (30 nM), HDAC4 (115 nM) HDAC6 (82 nM), HDAC7 (67 nM), HDAC8 (216 nM)	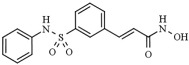	[91,92,93]
	Panobinostat	HDAC1 (3 nM), HDAC2 (3 nM), HDC3 (4 nM), HDAC4 (23 nM), HDAC6 (3 nM), HDAC7 (18 nM), HDAC8 (248 nM),	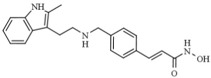	[94,95]
	Ricolinostat	HDAC1 (58 nM), HDAC2 (48 nM), HDAC3 (51 nM), HDAC6 (5 nM), HDAC8 (100 nM)	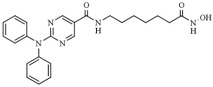	[104]
	Citarinostat	HDAC1 (35 nM), HDAC2 (45 nM), HDAC3 (46 nM), HDAC6 (3 nM), HDAC8 (137 nM)	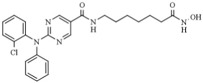	[110]
Benzamides	Entinostat	HDAC1 (190 nM), HDAC2 (650 nM), HDC3 (600 nM)	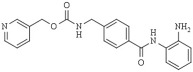	[115,116,117]
	Chidamide	HDAC1 (95 nM), HDAC2 (169 nM), HDAC3 (67 nM), HDAC10 (78 nM)	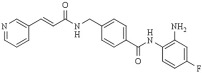	[118,119,120]
	MPT0L184	HDAC1 (90 nM), HDAC2 (400 nM), HDAC3 (2,3 μM)	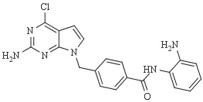	[119]
	Mocetinostat	HDAC1 (9 nM), HDAC2 (34 nM), HDAC3 (265 nM)	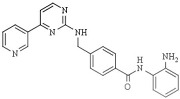	[122]
Cyclic peptides	Romidepsin	HDAC1, 3, -8 (<1 nM), HDAC4 (20 nM), HDAC6 (9 nM)	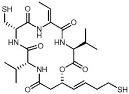	[48,127,128,129,130]

**Table 3 pharmaceutics-14-00209-t003:** Inhibition of HDAC activity by selected natural products in cancer cells.

Natural HDAC Inhibitors
Compound Name and Structure	Source	HDAC Isoforms	Ref.
Organosulfurs	allyl mercaptan 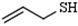	organosulfur compounds from garlic	HDAC 8	[154]
diallyl disulfide 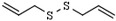
Isothiocyanates	Benzyl isothiocyanate 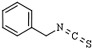	brassica or cruciferous vegetables	HDAC1 and 3	[160]
sulforaphane 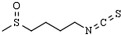	-HDAC1, -4, -6 and -7	[161,162]
Flavonoids	Quercetin 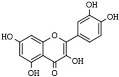	plants and fruits	HDAC1, HDAC8	[173,174,175]
Apigenin 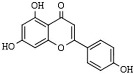	Asteraceae family	HDAC1 and -3	[177,178]
Chrysin 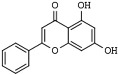	mushrooms, olive oil, tea, red wine, and passion fruit flowers, as well as Thai propolis and honey	HDAC-2, 3 and 8	[181]
Polyphenols	Curcumin 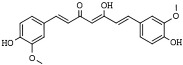	Curcuma longa	HDAC1, -3, -4, -6 and -8	[185,186,187]
(−)Epigallocatechin-3-gallate (EGCG) 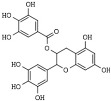	curry spices, grapes, soy, and berries	HDAC1, -2, and -3	[189,190,191,192]
Resveratrol 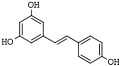	grapes and wine	pan-inhibitor	[195,196]
Isoflavone	Genistein 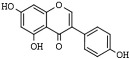	soybeans	HDAC1, -5 and -6	[199,200,201,202,203]

**Table 4 pharmaceutics-14-00209-t004:** Selected bifunctional HDAC inhibitors. Red shows HDAC pharmacophoric features, and blue represents receptor tyrosine kinase inhibitor (1), colchicine (2), JAK2 inhibitor (3), and lysine-specific demethylase 1 (LSD1) inhibitor (4).

Bifunctional HDAC Inhibitors			
Name	Targets (IC_50_)	Structure	Ref.
(1) CUDC-101	HDAC (4.2 nM) EGFR (2.4 nM) HER2 (15.7 nM)	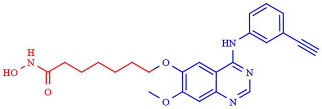	[233]
(2) 6a	HDAC1 (1.5 μM) HDAC2 (0.19 μM) HDAC3 (1.49 μM	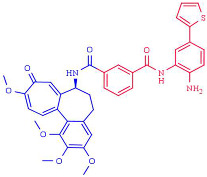	[237]
(3) JAK/HDAC6 dual inhibitor	HDAC6 (2.1 nM) JAK2 (1.4 nM) HDAC3 (2.17 μM)	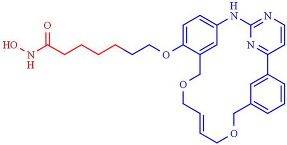	[238]
(4) LSD1/HDAC dual inhibitor	HDAC1 (15 nM) HDAC2 (23 nM) LSD1 (1.2 μM)	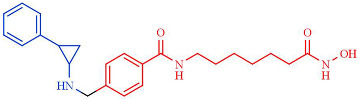	[240]

**Table 5 pharmaceutics-14-00209-t005:** Selected proteolysis targeting chimeras for HDACs degradation. Red shows HDAC pharmacophoric features, black shows linker regions, and blue shows E3 ligases. VHL: Von Hippel–Lindau; DC_50_, concentration to reduce the target protein expression in a 50%; DC_max_: maximum percentage of degradation; ND: non-determined.

Proteolysis Targeting Chimeras
Name	Targets (DC_50_, D_max_)	Structure	Ref.
9c—HDAC6 degrader	HDAC6(DC_50_ = 34 nM, D_max_ = 70.5%)	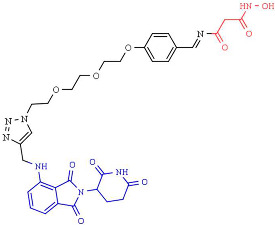	[248]
NP8	HDAC6(DC_50_ = 3.8nM, D_max_ = ND)	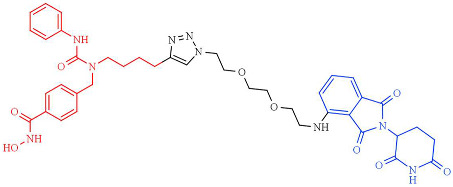	[249]
NH2	HDAC6(DC_50_ = 3.2 nM, D_max_ = ND)		[250]
VHL-Next-A degrader	HDAC6(DC_50_ = 7.1 nM, D_max_ = 90%)	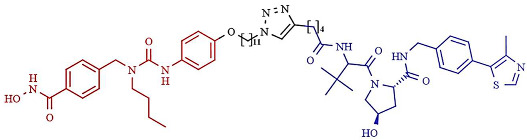	[251]
XZ9002 degrader	HDAC3(DC_50_ = 42 nM, D_max_ = 70%)	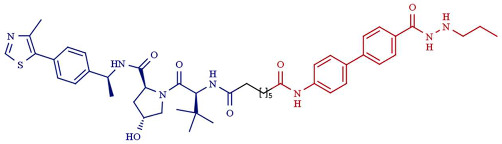	[252]
P1	HDAC1, HDAC6, and HDAC8	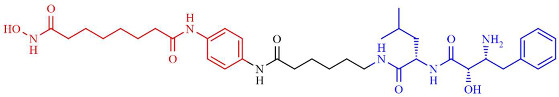	[254]

**Table 6 pharmaceutics-14-00209-t006:** Selected histone deacetylase inhibitors in combined clinical trials. TKi: tyrosine kinase inhibitors; PD-1; programed cell death receptor-1.

Combined Clinical Trails Strategies			
HDACi	Combined Targeting	Cancer Type	Ref.
Vorinostat	Phase I/II, gefitinib- EGFR-TKi	Non-small-cell lung cancer	[262]
Vorinostat	Phase II, bevacizumab- angiogenic VEGF blocker	Metastatic clear-cell renal cell carcinoma	[264]
Tucidinostat	Phase III, exemestane-steroidal aromatase inhibitor, hormonal therapies	Hormone receptor-positive (HR^+^) and HER2 negative breast cancer	[266]
Entinostat	Phase I, testosterone antagonist therapy-enzalutamide, hormonal therapy	Castration-resistant prostate cancer	[267]
Entinostat	Placebo-controlled phase III study, exemestane-steroidal aromatase inhibitor, hormonal therapy	Hormone receptor-positive (HR^+^) and HER2-negative breast cancer	[269]
Panobinostat	Phase I dose-finding trial, -mTOR inhibitor-everolimus, autophagy	Advanced clear-cell renal cell carcinoma	[271]
Tucidinostat	Phase II, cisplatin, chemotherapy	Triple-negative breast cancer	[274]
Romidepsin	Phase I dose-escalation study, liposomal doxorubicin chemotherapy	Cutaneous T-cell lymphoma	[275]
Mocetinostat	Non-randomized phase I/II, gemcitabine chemotherapy	Various solid tumors, including advanced pancreatic cancer	[277]
Vorinostat	Phase II, immune checkpoint inhibitor anti–PD-1-pembrolizumab, immunotherapy	Recurrent/metastatic squamous cell carcinomas of the head and neck and salivary gland cancer	[280]
Entinostat	Phase II, immune checkpoint inhibitor anti–PD-1-pembrolizumab, immunotherapy	Metastatic uveal melanoma	[281]
Citarinostat	Phase Ib, immune checkpoint inhibitor anti–PD-1-nivolumab, immunotherapy	Non-small-cell lung cancer	[282]
Romidepsin	Phase I, immunomodulatory drug lenalidomide and the proteasome inhibitor carfilzomib	T-cell lymphoma and B-cell lymphoma	[283]

## Data Availability

The material supporting the conclusion of this review has been included in the article.

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
