# Peer review of "Targeting Histone Deacetylases: Opportunities for Cancer Treatment and Chemoprevention"

_pharmaceutics, 2022, doi:10.3390/pharmaceutics14010209_

Round 1

Reviewer 1 Report

The authors introduced the histone deacetylation and the inhibitor molecules for this process. Also, this review also concluded the HDACi features' molecular structures, mechanisms, and functions as potential anticancer drug candidates.  Furthermore, the ideas to design new molecules are mentioned the achieve more isoform-selective inhibitors. This review provides a comprehensive view of recent HDACi research, it is meaningful to the future HDACi study. I suggest accepting this review paper.

Author Response

Reviewer #1:

Comments and Suggestions for Authors

The authors introduced the histone deacetylation and the inhibitor molecules for this process. Also, this review also concluded the HDACi features' molecular structures, mechanisms, and functions as potential anticancer drug candidates. Furthermore, the ideas to design new molecules are mentioned the achieve more isoform-selective inhibitors. This review provides a comprehensive view of recent HDACi research, it is meaningful to the future HDACi study. I suggest accepting this review paper.

Answer: We appreciate the positive comments of reviewer #1 to the manuscript.

Reviewer 2 Report

none

Author Response

Reviewer #2:

Comments and Suggestions for Authors

None

Answer: We thank reviewer #2 for the report of our review manuscript.

Reviewer 3 Report

The study aims to describe first generation of classic small molecule and dietary-derived compounds HDAC inhibitors and their anticancer and chemopreventive activities. Moreover, the authors  discuss some perspectives of addressing new ways of improving the HDACi selectivity and the combinatory potential of HDACi, and other specific targets that could be included in future clinical applications.

Several small molecule HDAC inhibitors have been synthetized and tested as anticancer agents, both in preclinical and clinical setting.

The review lacks originality, many aspects of HDACi have already been described, such as the table of drugs, or the natural compounds targeting histone deacetylases. Authors should summarize the first part and expand the role of hybrid compounds or proteolysis targeting chimeras.

The review is good from a pharmaceutical point of view but not from a clinical point of view.

In summary although the topic is interesting the manuscript is unacceptable in this current form, the study looks promising but I would like to encourage the authors to consider a deeper review.

Author Response

REVIEWER # 3

We thank reviewer #3 for the detailed and valuable comments, observations, and suggestions. This will definitely increase the credentials of this manuscript. Kindly find below our point-wise responses, and the same has been highlighted in the modified manuscript.

Comments and Suggestions for Authors

The study aims to describe first generation of classic small molecule and dietary-derived compounds HDAC inhibitors and their anticancer and chemopreventive activities. Moreover, the authors discuss some perspectives of addressing new ways of improving the HDACi selectivity and the combinatory potential of HDACi, and other specific targets that could be included in future clinical applications.

Several small molecule HDAC inhibitors have been synthetized and tested as anticancer agents, both in preclinical and clinical setting.

Specific comments:

-The review lacks originality, many aspects of HDACi have already been described, such as the table of drugs, or the natural compounds targeting histone deacetylases.

Authors should summarize the first part and expand the role of hybrid compounds or proteolysis targeting chimeras.

Answer: We understand the reviewer's concern. There are many excellent reviews on the medicinal chemistry of HDAC inhibitors, their preclinical development, and current clinical applications (that we cited). Most review publications on HDAC inhibitors, whether in Pharmaceutics or elsewhere, focus on their anticancer properties. We have gone a significant step further and comparatively discussed the chemopreventive and therapeutic potential of various natural and synthetic HDAC inhibitors. Also, we presented the Pharmaceutics readership with the molecular mechanisms of HDAC inhibition, their role in cancer biology, and differences between chemopreventive and chemotherapeutic HDAC inhibitors in chemistry and pharmacology.

We believe that, although some information may lack novelty, the data used in the first part of this review may be interesting for Pharmaceutics' readerships, and we would like to keep it in the current form to maintain the main essence of the manuscript.

Also, as is suggested by the reviewer, we expanded and made a reorganization according to E3 ligase binding motifs of  section  4.3) on Proteolysis targeting chimeras (the added parts are labeled in blue)

-The review is good from a pharmaceutical point of view but not from a clinical point of view.

Answer: We appreciate the reviewer comments; according to that, we try to fit our review with the main scopes of the journal Pharmaceutics; nevertheless, we agree with the referee that a clinical point of view may increase the medical sound of the manuscript. Accordingly, we have added a new section 5—combinatory clinical perspectives for histone deacetylases inhibitors (text is labeled in violet).

-In summary although the topic is interesting the manuscript is unacceptable in this current form, the study looks promising, but I would like to encourage the authors to consider a deeper review.

Answer: To the best of our knowledge, this is the unique review that compares in vitro and in vivo effects of known natural and synthetic HDAC inhibitors, already tested in clinical trials as chemopreventive and chemotherapeutic agents. Basic research covered in this review (discovery of HDACs, medicinal chemistry of HDAC inhibitors, chemical biology of HDAC inhibition) and pharmaceutical-to-clinical application (bifunctional HDACi, PROTAC HDACi, and clinical perspectives of HDACi) portray our most recent understanding of the HDACi in oncology.

Reviewer 4 Report

The manuscript quoted pharmaceutics-1508022 "Targeting Histone Deacetylases: Opportunities for Cancer Treatment and Chemoprevention" was submitted to Pharmaceutics as review.

The manuscript describes the role of HDAC inhibitors for cancer treatment. the introduction section is very well organized, offering the possibility to appreciate the 11 HDAC isofoms. Nevertheless, when introducing the focus of the review it is somewhat confusing and should be restyled/rephrased according with the following suggestions.

The specific role of different HDACs must be highlighted describing the different isoforms and their inhibitors. Another strategy could be the discussion of different tumor chemotypes in association to the expression of HDAC isoform in the tissue.

Data from clinical trials must be inserted to discover the recent findings in cancer research.

The paragraph related to the PROTAC is nicely incorporated, but it should be contextualized as a translational use of HDACis.

A particular emphasis on HDAC1 and HDAC6 must be highlighted, considering their predominant role in cancer progression.

Specific and recent literature data must be inserted: https://doi.org/10.1021/acsmedchemlett.0c00395,

 https://doi.org/10.1016/j.ejmech.2020.112998

Author Response

Reviewer #4:

Comments and Suggestions for Authors

The manuscript quoted pharmaceutics-1508022 "Targeting Histone Deacetylases: Opportunities for Cancer Treatment and Chemoprevention" was submitted to Pharmaceutics as review.

The manuscript describes the role of HDAC inhibitors for cancer treatment. the introduction section is very well organized, offering the possibility to appreciate the 11 HDAC isofoms. Nevertheless, when introducing the focus of the review it is somewhat confusing and should be restyled/rephrased according with the following suggestions.

1-The specific role of different HDACs must be highlighted describing the different isoforms and their inhibitors. Another strategy could be the discussion of different tumor chemotypes in association to the expression of HDAC isoform in the tissue.

Answer:

We appreciate the reviewer's comments and suggestions that help to improve our review manuscript.

We conduct the review from the view of the HDACi inhibitors structures to highlight their results in cancer. Many of the HDACi included in the manuscript can inhibit more than one of the isoforms; therefore,  we thought it would be more feasible to expose the literature data for HDACi in cancer,  according to the HDACi's ZBG classification. When we started with the review writing, we explored some other ways, as the reviewer point-out, to define the discussion of HDACi in cancer, but finally, and according to the scope of the journal Pharmaceutics, we decided that maybe it is better to expose the  HDACis from a structural point of view, and also similarly for dietary-derived HDACi.

2-Data from clinical trials must be inserted to discover the recent findings in cancer research.

Answer: We thank the reviewer's suggestions, and, accordingly, we have included a new section of clinical trials with combined HDACi with some current onco-therapies (Violet text), mainly in solid tumors. Also, several valuable reviews have already analyzed HDACi clinical trials as monotherapy, and recently the use of HDACi in combined therapies seems more feasible for future cancer treatments.

We hope this new section can give a more clinical perspective of the use of HDACi in combined potential cancer treatments.

3-The paragraph related to the PROTAC is nicely incorporated, but it should be contextualized as a translational use of HDACis.

Answer: We are grateful for the valuable reviewer suggestion. As is recommended, we have added some paragraphs indicating the post-translational function of PROTACS in the knockdown of the protein of interest (Green text)

4-A particular emphasis on HDAC1 and HDAC6 must be highlighted, considering their predominant role in cancer progression.

Answer: We understand the reviewer's concern and agree with the importance of HDAC1 and HDAC6 in tumorigenesis. Although we indicated the importance of the development of HDACis with enhanced HDACs specificity for the cancer treatment,   we would like to keep some balance of the targeting of HDACs in cancer according to the relatively low HDACs specificity of the HDACis approved for cancer therapy and the others included in this review.

We think that the role of HDAC1 and HDAC6 in cancer merit a focalized review that we would like to address in the future.

Round 2

Reviewer 3 Report

The manuscript has been improved as requested even if the first part has remained unchanged but I understand the intentions of the authors.